# Bioproduction Optimization, Characterization, and Bioactivity of Extracellular Pigment Produced by *Streptomyces parvulus*

**DOI:** 10.3390/ijms262110762

**Published:** 2025-11-05

**Authors:** Laura Daniela Silva-Arias, Luis Díaz, Ericsson Coy-Barrera

**Affiliations:** 1Master’s Program in Process Design and Management, School of Engineering, Universidad de La Sabana, Chia 140013, Colombia; laurasiar@unisabana.edu.co; 2Bioprospecting Research Group, School of Engineering, Universidad de La Sabana, Chia 140013, Colombia; 3Agroindustrial Production Research Group, Doctorate Program in Biosciences, School of Engineering, Universidad de La Sabana, Chia 140013, Colombia; 4Bioorganic Chemistry Laboratory, Universidad Militar Nueva Granada, Cajicá 250247, Colombia; ericsson.coy@unimilitar.edu.co

**Keywords:** actinomycetes, pigment, antityrosinase, optimization, anti-inflammatory, anti-acne, cosmeceutical

## Abstract

Actinomycetes, especially *Streptomyces*, are prolific producers of bioactive metabolites, including pigments with potential applications in foods, textiles, cosmetics, and pharmaceuticals. Motivated by increasing concerns about the safety and environmental impact of synthetic pigments, this study aimed to optimize the production of an extracellular pigment-rich fraction from *Streptomyces parvulus* and to evaluate its bioactivities relevant for cosmeceuticals. A Plackett–Burman design was used to identify key variables influencing metabolite production, followed by optimization with a Box–Behnken design. The pigment-rich fraction was obtained after extraction with ethyl acetate from lyophilized supernatants and chemically characterized by IR and LC–MS. Biological assays were conducted to assess anti-tyrosinase, immunomodulatory, and antimicrobial activities. Temperature, incubation time, and agitation speed were identified as the most significant factors, with optimal conditions of 30 °C, 50 rpm, and 7 days yielding a pigment concentration of 465.3 μg/mL. LC–MS analysis revealed three 1,4-naphthoquinone-containing compounds, annotated as juglomycin Z (**1**), WS-5995B (**2**), and naphthopyranomycin (**3**), as the main constituents. The pigment-rich fraction showed modest anti-tyrosinase activity (10.9% at 300 μg/mL), immunomodulatory effects (TNF-α inhibition up to 36.9% and IL-10 stimulation up to 38.4% in macrophages), and antimicrobial activity against *Staphylococcus epidermidis* (15.8 mm inhibition halo, 91% growth reduction). The optimized fermentation model enhances pigment yield while reducing resource consumption, and the pigment-rich fraction exhibits multifunctional bioactivities, underscoring its potential as a natural cosmeceutical ingredient.

## 1. Introduction

The genus *Streptomyces*, belonging to the order Actinomycetales, has attracted extensive scientific and industrial attention due to its remarkable metabolic diversity [1]. More than 76% of all known bioactive secondary metabolites are derived from actinomycetes, with *Streptomyces* species producing a broad range of antimicrobial, anticancer, antioxidant, and pigment-forming compounds [2,3]. These metabolites have driven significant advances in pharmaceuticals, agriculture, and cosmeceuticals.

In recent years, increasing consumer demand for natural, sustainable ingredients has renewed interest in *Streptomyces*-derived pigments as alternatives to synthetic dyes in high-value sectors [4,5]. Beyond their coloring capacity, microbial pigments often exhibit functional properties such as antioxidant, antimicrobial, and anti-inflammatory activities, which enhance their appeal for use in skincare and other health-related applications [6,7,8,9,10]. However, large-scale pigment production by microorganisms is limited by challenges, including low yields, high production costs, and inefficient extraction processes [11]. Optimizing culture conditions is therefore essential to improve pigment yield, reduce costs, and support the evaluation of biological activities. In addition to optimizing culture conditions traditionally used to enhance metabolite production, recent research has increasingly emphasized the genetic modification of *Streptomyces* through metabolic engineering and synthetic biology. These cutting-edge strategies enable the redesign and activation of silent biosynthetic gene clusters (BGCs), fine-tuning of regulatory pathways, and construction of strains with improved metabolic efficiency and productivity. The development of advanced molecular tools—such as CRISPR/Cas systems, integrative and replicative plasmids, synthetic promoters, and modular vector architectures—has greatly enhanced the precision and efficiency of *Streptomyces* engineering. As a result, the discovery and large-scale production of novel bioactive compounds have been markedly accelerated [12]. As highlighted by Krysenko [12], these innovations are driving a shift from empirical strain improvement toward rational, design-based optimization, positioning *Streptomyces* as one of the most versatile and powerful microbial platforms for modern biotechnological applications.

Microbial pigments represent an expanding field of study, not only for their vibrant chromophores but also for their structural complexity and bioactive potential. Compounds such as actinorhodin, prodigiosin analogues, and melanin-like molecules from *Streptomyces* species have been explored for therapeutic, antimicrobial, and industrial uses [13,14,15]. Despite these advances, relatively few strains have been developed for scalable pigment production, and most studies focus on model organisms, such as *S. coelicolor* or *S. griseus* [16,17]. This fact highlights the importance of characterizing lesser-studied species as potential sources of novel pigments with distinctive properties. Among these, *Streptomyces parvulus* remains underexplored compared to other species in the genus. Although it is recognized as a producer of antibiotics such as actinomycin D [15], there is limited information on its pigment biosynthetic capacity or the structural and functional properties of these metabolites. Characterizing pigments from *S. parvulus* not only expands the chemical space of microbial colorants but also provides an opportunity to discover multifunctional compounds with applications in emerging fields such as cosmeceuticals.

The cosmeceutical sector is rapidly growing, with global market projections exceeding USD 70 billion by 2028, driven by consumer demand for products that offer dual functionality—both aesthetic and therapeutic [18]. Pigments that combine natural coloration with antioxidant, anti-inflammatory, or antimicrobial properties directly address this demand, particularly in skincare products targeting hyperpigmentation, oxidative stress, or acne-related disorders. Current options rely heavily on synthetic additives or plant-derived extracts, which have variable yields and stability, underscoring the need for reliable, microbial-based pigment sources.

The ecological origin of microbial strains also plays a key role in their metabolic repertoire. Colombia’s Arauca River basin, characterized by tropical biodiversity and unique physicochemical conditions, represents an underexplored niche for isolating microorganisms with distinctive biosynthetic capabilities [19,20]. Strains from such environments may harbor unusual gene clusters encoding specialized metabolites not found in better-studied species, providing a valuable reservoir for natural product discovery. Thus, the isolation of a pigment-producing *S. parvulus* strain from this habitat underscores both the novelty and biotechnological potential of the present work.

Building on our preliminary findings showing that this strain produces extracellular pigments with potential cosmetic relevance [19], the objectives of the present study were to: (i) optimize pigment production through factorial experimental design; (ii) characterize the pigment-rich fraction by FTIR and LC–MS; and (iii) assess its bioactivities, including anti-tyrosinase, anti-inflammatory, and anti-acne properties. This combined optimization and characterization approach highlights the potential of *S. parvulus*-derived pigments as multifunctional, natural ingredients for cosmeceutical applications.

## 2. Results

### 2.1. Selected Streptomyces Strain S145 and Response Variable

From an in-house Actinobacteria collection obtained from Arauca River sediments, a strain with *Streptomyces*-like morphology, cultivated and preserved in ISP2 medium, was selected [19]. In previous studies by our group, this strain (S145) was selected and exhibited intense yellow pigmentation in ISP2 culture medium. Identification by fp sequencing revealed 98.7% similarity to *S. parvulus*. Accordingly, strain S145, the producer of the pigment in this study, was assigned to *S. parvulus* S145.

Pigment production was evaluated by measuring the yield of a pigment-rich fraction obtained by ethyl acetate extraction of lyophilized culture supernatants. Initial assessments were performed under the reported fermentation conditions, followed by modified conditions as part of the optimization process. The pigment yield, expressed as µg/mL using the corresponding standard curve (Materials and Methods, Section 4.1.2), was adopted as the response variable for selection and optimization.

### 2.2. Evaluation of the Main Factors That Influence Pigment Production

#### 2.2.1. Carbon Source Selection

Five carbon sources (glucose, starch, lactose, glycerol, and sucrose) were evaluated under solid-state fermentation at 30 °C for seven days. As shown in Figure 1, soluble starch had the most pronounced effect on pigment production by strain S145, resulting in an intense yellow–orange coloration within seven days. Colonies cultivated with soluble starch exhibited early growth, with visible development by day 3, followed by fully matured, pigmented colonies by day 5. The photographs were taken against a black background to enhance contrast, highlighting the slightly brownish coloration of the plate containing soluble starch. This pigmentation corresponds to the desired compound, selected because the strain was cultivated in ISP-2 medium. The blue arrow in Figure 1A indicates the yellow pigment of interest that was subsequently analyzed. In contrast, sucrose promoted poor, irregular growth, with delayed colony development and reduced pigmentation. Glucose, lactose, and glycerol supported moderate growth, resulting in uniform colonies by day 5 and complete maturation by day 7, although with less intense pigmentation than starch. The negative control (ISP2 medium without inoculation) showed no visible changes.

#### 2.2.2. Nitrogen Source Selection

Five nitrogen sources (ammonium sulfate, potassium nitrate, ammonium chloride, yeast extract–malt extract, and casein) were assessed under the same culture conditions. Pigment production was most effectively supported by yeast extract–malt extract, the standard components of ISP2 medium. As shown in Figure 2, growth was completely absent in media supplemented with inorganic nitrogen sources (potassium nitrate, ammonium sulfate, and ammonium chloride). In contrast, casein and yeast extract–malt extract both supported growth and pigment production. Yeast extract–malt extract led to more rapid growth, with visible colony development by day 3 and abundant pigmentation by day 5. Casein also supported pigment production, but with a delay of approximately one day compared to yeast–malt. From a practical perspective, yeast extract–malt extract was preferred for its growth-promoting efficiency and cost-effectiveness compared to casein. The photographs were taken against a black background to enhance contrast, revealing the slightly brownish coloration of the plates containing casein or yeast extract–malt extract. This pigmentation corresponds to the desired compound, produced by the strain cultivated in ISP-2 medium. The blue arrow in Figure 2C indicates the yellow pigment selected for further analysis.

#### 2.2.3. Plackett–Burman Experimental Design

The selected carbon source (soluble starch) and nitrogen source (yeast and malt extracts) were used to evaluate the main factors influencing pigment production through a Plackett–Burman design. Twelve runs were performed to assess the effects of temperature, pH, carbon source concentration, nitrogen source concentration, agitation, and fermentation time (see Materials and Methods, Section 4.2). Pigment production varied significantly across experimental conditions, as indicated by the Plackett–Burman results (Table 1).

The lowest yield (0.07 μg/mL) was obtained under the following conditions: 25 °C, pH 6, 3 days of incubation, 0.1% carbon and nitrogen concentration, and 100 rpm. In contrast, the highest yield (85.33 μg/mL) was achieved at 30 °C, pH 7, seven days of incubation, 0.3% carbon, 0.1% nitrogen, and 50 rpm. These findings suggest that temperature, pH, agitation speed, and incubation time have a significant impact on pigment production, whereas nitrogen concentration has a minimal effect on this process.

Statistical analysis confirmed this observation. The regression model showed a strong correlation between independent factors and pigment concentration (R^2^ = 96.66%, adjusted R^2^ = 92.65%, predicted R^2^ = 80.77%). Among the evaluated variables, temperature, pH, agitation speed, and incubation time were statistically significant (*p* < 0.05), whereas nitrogen concentration was not statistically significant (*p* > 0.05). The Pareto chart (Figure 3) illustrates the relative influence of each factor.

By excluding the non-significant terms from the model, the resulting first-order polynomial equation describes the production of yellow–orange extracellular pigment by *Streptomyces* strain S145 as a function of the independent variables. The first-order polynomial model was defined as (Equation (1)):(1)Y=39.66+11.50Tem−15.17pH+10.73Ti+9.48CC+16.94rpm
where *Y* is pigment concentration (µg/mL), Tem = temperature (°C), pH = culture pH, Ti = incubation time (days), CC = carbon source concentration (%), and rpm = agitation speed.

#### 2.2.4. Response Surface Methodology (Box–Behnken Design)

Based on the previously mentioned first-order polynomial equation, the model was optimized using a Box–Behnken response surface design, focusing on the factors with the highest significance or positive effects on the response variable. As indicated by the preceding statistical analyses and the fitted model, the most influential factors were temperature, agitation time, and agitation speed. Using these three variables, fifteen experimental runs were performed in biological triplicate (see Materials and Methods, Section 4.3), along with their respective coding and pigment production values. Based on the Plackett–Burman results presented in Table 2, the relationship between the dependent and independent factors was established to determine the maximum production of the yellow-orange pigment. Consequently, the second-order polynomial model was defined as (Equation (2)):(2)$$Y=227.28-201.51A-71.30B+115.35AB-150.88B^{2}$$
where *Y* is pigment concentration (µg/mL), *A* = temperature (°C), and *B* = agitation speed (rpm), incubation time was excluded from the model due to its lack of significance in the interaction terms.

ANOVA analysis (Table 2) demonstrated that the model was significant (*p* < 0.0001) and showed a high correlation (R^2^ = 0.9006, adjusted R^2^ = 0.8609, predicted R^2^ = 0.7914), indicating good model predictability and reliability (predicted R^2^ > 0.7). Temperature and agitation speed showed the most decisive influence, including their interaction term (AB) and quadratic effect (B^2^).

The response surface plot and contour map illustrate the interactive effects of temperature and agitation. Maximum pigment production was achieved at 30 °C, 50 rpm, and 7 days of incubation, yielding 465.55 μg/mL. In contrast, higher temperature (37 °C) and faster agitation (113 rpm) reduced pigment yield to 189.09 μg/mL. In this regard, optimization led to a 5.5-fold increase in pigment production (Figure 4A,B). The pigment production response surface at different design points revealed the optimal ranges of incubation time, temperature, and agitation for maximizing pigment yield (Figure 4C). The quality of the optimized model was supported by the standard probability plot of residuals (Figure 4D), which indicated strong confidence in its predictive accuracy. The model was further validated in a 1 L bioreactor under optimal conditions, yielding 457 μg/mL with a model error of only 1.72%, thereby confirming the accuracy and reliability of the statistical optimization.

### 2.3. Chemical Characterization of the S. parvulus S145-Derived Pigment-Rich Fraction

The UV–Vis absorption spectrum (Appendix A) of the pigment-rich fraction from the optimized *S. parvulus* S145 culture showed a strong maximum below 350 nm, with a secondary shoulder at 410–470 nm. Absorbance rapidly decreased above 480 nm, with negligible signals beyond 500 nm. This profile indicates predominant absorption in the near-UV and violet–blue regions, consistent with conjugated aromatic chromophores, such as polyphenolic or polyketide-derived structures, commonly reported in *Streptomyces*, which may still impart a light yellow to orange hue [21]. The FTIR spectrum (Appendix A) further supported this assignment, revealing a broad O–H stretch at 3400–3200 cm^−1^ (phenolic groups), C–H stretching bands at 2925–2850 cm^−1^ (aliphatic side chains), an intense C=O stretch at 1630–1650 cm^−1^ (quinone or conjugated carbonyls), and aromatic C=C stretches at 1600–1500 cm^−1^. These features confirm the presence of phenolic hydroxyls, conjugated carbonyls, and aromatic systems, consistent with an aromatic polyketide scaffold of the naphthoquinone type. Complementary LC–MS analysis (Appendix A) identified three major metabolites, annotated at level 2 using the StreptomeDB 4.0 database [20] based on HRMS spectra acquired in negative ion mode (Appendix A). These features and their respective pattern fragmentation were assigned as juglomycin Z (1) (*m*/*z* 289.0721), WS-5995B (2) (*m*/*z* 337.0719), and naphthopyranomycin (3) (*m*/*z* 471.1655), all with mass accuracy errors of less than 5 ppm. Together, the UV–Vis, FTIR, and LC–MS data strongly support the chemical composition of the pigment-rich fraction as a three-component mixture of yellow–orange 1,4-naphthoquinone-like polyketides, in agreement with typical pigment profiles of *Streptomyces* species [22,23,24].

### 2.4. Bioactivity of S. parvulus S145-Derived Pigment-Rich Fraction

#### 2.4.1. Tyrosinase Inhibitory Activity

The inhibitory effect of the *S. parvulus* S145-derived pigment-rich fraction on tyrosinase was compared with that of kojic acid, a standard inhibitor (Figure 5). Kojic acid consistently displayed higher inhibition across all tested concentrations (51.85–89.63% at 50–1000 µg/mL). In contrast, the pigmented fraction showed a concentration-dependent response, reaching its maximum inhibitory activity of 29.63% at 1000 µg/mL. Statistically significant differences (*p* < 0.05) were observed among the tested concentrations (10.37% at 30 µg/mL; 20.74% at 300 µg/mL). These results demonstrate that, while less potent than kojic acid, the pigment fraction exhibits moderate tyrosinase inhibitory activity, suggesting potential use as a natural skin-lightening or anti-browning agent.

#### 2.4.2. Anti-Inflammatory Activity

THP-1 monocytes (MCs) were differentiated into macrophages (MFs) by incubation with phorbol 12-myristate 13-acetate (PMA) (50 ng/mL, 24 h). Both MCs and MFs expressed comparable tumor necrosis factor-alpha (TNF-α) levels, although MFs exhibited a 3–5-fold increase in interleukin-10 (IL-10) expression. Upon Lipopolysaccharide (LPS) stimulation (50 ng/mL), MFs displayed a 100-fold rise in TNF-α expression, confirming the induction of an inflammatory phenotype. Treatment with *S. parvulus* S145-derived extract (MFLP-Ex, 300 µg/mL) and the pigmented fraction (MFLP-Fr, 300 µg/mL) significantly reduced TNF-α secretion by 23.24% and 36.89%, respectively, compared to LPS-stimulated macrophages (*p* < 0.05). The effect of MFLP-Fr was comparable to that of ibuprofen (Ib) (50 µg/mL) (Figure 6A).

In terms of anti-inflammatory cytokines, IL-10 expression increased by 23.84% (MFLP-Ex) and 38.42% (MFLP-Fr) compared to the LPS control, whereas ibuprofen did not induce a similar effect (Figure 6B). These findings suggest that the *S. parvulus* pigment fraction exerts dual immunomodulatory effects by downregulating the proinflammatory cytokine TNF-α and upregulating the anti-inflammatory cytokine IL-10.

#### 2.4.3. Anti-Acne Activity

The antimicrobial potential of the pigment fraction was evaluated against *Staphylococcus epidermidis*, a major opportunistic skin pathogen. In disk diffusion assays, the *S. parvulus* pigmented fraction at 10 mg/mL produced an inhibition halo of 17.8 mm, surpassing the effect of the positive control, i.e., vancomycin (8.0 mm) (Figure 7A). The water–ethanol vehicle control showed no inhibitory activity.

Growth kinetics confirmed these results (Figure 7B). At 300 µg/mL, the pigmented fraction inhibited 91% of bacterial growth, with the growth curve fitting a linear model (µ = 0.00917 ± 0.000749 h^−1^). At lower concentrations (30 and 3 µg/mL), no significant inhibition was observed, and the growth curves were comparable to the control. Vancomycin (0.4–40 µg/mL) effectively inhibited *S. epidermidis* growth by 96–98%, showing a more substantial effect than the pigment fraction. Interestingly, the fraction at 300 µg/mL exhibited a bacteriostatic effect, delaying bacterial proliferation for up to 12 h, after which growth resumed. Together, these results suggest that the *S. parvulus* pigment fraction possesses reasonable antimicrobial activity, particularly against acne-associated bacteria, and could serve as a natural candidate for topical formulations.

#### 2.4.4. Cytotoxic Activity

Cytotoxic activity was assessed in human HDFa (human dermal fibroblast adult) and HaCaT (human keratinocyte) cell lines using the 3-(4,5-dimethylthiazol-2-yl)-2,5-diphenyltetrazolium bromide (MTT) assay. Both the crude extract and the pigmented fraction caused a significant reduction (*p* < 0.001) in cell viability in both cell lines at 300 µg/mL (Figure 8), compared with untreated controls. The extract reduced cell viability to 82.0% in HDFa and 87.5% in HaCaT cells, while the pigmented fraction yielded viabilities of 81.3% and 92.7%, respectively. In contrast, cells treated with fetal bovine serum (FBS, 10%) exhibited enhanced proliferation, reaching viability levels of 138.6% in HDFa and 128.3% in HaCaT cells, indicating a more substantial mitogenic effect in fibroblasts after 24 h of treatment. Dimethyl sulfoxide (DMSO) at concentrations of 5% and 10% served as positive cytotoxicity controls, resulting in viability values of 35.1% and 22.2% in HDFa and 46.8% and 25.1% in HaCaT cells, demonstrating a higher cytotoxic effect in the fibroblast line.

#### 2.4.5. Stability Studies of the Pigment-Rich Fraction from *S. parvulus*

The effect of light exposure on pigment stability was most pronounced during the first two hours, when the residual rate decreased to 71.62 ± 1.39%. Beyond this point, the rate remained relatively constant up to 12 h of exposure (71.15 ± 2.19%), as the maximum duration evaluated (Figure 9A). In contrast, samples kept in darkness maintained or even increased their residual rate (111.68 ± 0.80%), indicating that light exposure promotes pigment degradation. The initial time point (0 h) served as the reference for all comparisons. The thermal stability assay revealed no significant differences (*p* < 0.0001) in the residual rate across temperatures ranging from 0 °C to 90 °C (Figure 9B), with the residual rate remaining close to 100% throughout the 2 h incubation. The reference temperature for comparison was 20 °C (room temperature). In the pH stability test, the residual rate remained around 100% between pH 5 and 7, increased under acidic conditions (pH 3: 109.39 ± 1.15%, pH 1: 112.38 ± 1.25%), and slightly under basic conditions (pH 9: 106.23 ± 1.82%). However, at highly basic pH values, the rate dropped sharply (pH 11: 86.57 ± 0.86%, pH 13: 36.41 ± 1.10%), with the latter showing a noticeable color change to pink (Figure 9C). All tests were conducted for 2 h in darkness, using pH 5—the initial pH of the pigmented fraction—as the reference treatment.

## 3. Discussion

Three distinct developmental stages characterize the growth cycle of *Streptomyces*. Initially, spores germinate, producing germ tubes that extend into a dense vegetative mycelium. This stage provides the foundation for aerial hyphae formation, which occurs in the second stage and is associated with intensive nutrient consumption. Finally, aerial hyphae mature into spore chains, completing the life cycle [25,26]. Importantly, this study examined how such developmental dynamics can be altered by modifying the culture medium composition, particularly the sources of carbon and nitrogen, which are central to bacterial metabolism and secondary metabolite biosynthesis [3,11].

Our findings revealed that soluble starch was the most favorable carbon source for pigment production, consistent with previous reports showing enhanced metabolite yields in *Streptomyces* cultured on starch-based media [2,11]. The slower hydrolysis of starch compared to glucose likely reduced catabolite repression, prolonging carbon availability and promoting secondary metabolism, including pigment biosynthesis. In contrast, variation in nitrogen sources had a minimal impact compared to the standard yeast–malt extract medium, which supported robust growth and pigment production. The cost-effectiveness of yeast–malt extract compared with casein further supports its suitability for scale-up processes, aligning with earlier observations in *Streptomyces kanasenisi* ZX0, where starch and yeast extract enhanced glycoprotein production [2].

Beyond nutrient availability, abiotic factors such as temperature, incubation time, and agitation significantly influenced pigment biosynthesis. The Plackett–Burman design identified agitation as the most significant variable, with strong statistical support (*p* = 0.001). The Box–Behnken optimization further highlighted the critical role of temperature and agitation in determining pigment yield. The model’s reliability was evidenced by a high coefficient of determination (R^2^ = 0.9006) and adequate predictive performance, as indicated by a slight difference between the predicted R^2^ (Pred-R^2^) and the Adjusted R^2^ (Adj-R^2^) of 0.07. Moreover, the low-cost optimal conditions (30 °C, 7 days, 50 rpm) provide an economically feasible framework for laboratory-scale production. The estimated production cost ($135.71/μg) suggests competitiveness compared to high-cost orange-yellow pigments [27], positioning the *S. parvulus*-derived pigment as a promising alternative for industrial applications.

The bioactivity evaluation of the pigmented fraction underscores its potential in cosmeceutical and dermatological applications. The pigment exhibited moderate but significant tyrosinase inhibition (29.6% at 1000 µg/mL), suggesting potential for skin depigmentation or anti-browning formulations. While kojic acid remains a more potent inhibitor, the pigment fraction demonstrated a more stable inhibitory profile under experimental conditions, potentially offering advantages in formulation stability. Given the regulatory limitations and safety concerns surrounding kojic acid [28,29], natural pigments with reduced cytotoxicity and greater stability represent attractive alternatives.

Regarding anti-inflammatory activity, the pigment fraction significantly reduced TNF-α secretion and increased IL-10 production in LPS-stimulated macrophages. This dual modulation is noteworthy, as it parallels the effects of ibuprofen while also promoting the upregulation of anti-inflammatory cytokines. Similar results have been reported for nitrogen-containing secondary metabolites from endophytic *Streptomyces* spp., which suppressed proinflammatory mediators such as TNF-α, IL-1α, IL-6, and PGE_2_ in macrophage models [30]. The concurrent reduction in TNF-α and the enhancement of IL-10 by the pigment fraction suggest the presence of structurally diverse metabolites with immunomodulatory properties, warranting further characterization.

Antimicrobial assays against *S. epidermidis*, a bacterium commonly associated with acne [31], revealed strong inhibitory effects at higher pigment concentrations. The pigment-rich fraction exhibited bacteriostatic activity, delaying bacterial growth kinetics and producing inhibition halos superior to those of vancomycin. Although vancomycin remained more potent, the pigment’s activity supports its potential application as a natural anti-acne agent, particularly within topical formulations.

For cytotoxicity assays, the HDFa and HaCaT cell lines were selected because they represent the two principal cell types of human skin, the primary target tissue for potential topical or biomedical applications of the pigment extract and its fraction [32]. HDFa fibroblasts play a critical role in extracellular matrix synthesis and wound repair [33]. In contrast, HaCaT keratinocytes provide a robust, well-characterized model of epidermal responses and barrier function [34]. Evaluating both cell types, therefore, offers complementary insights into dermal compatibility and epithelial tolerance.

The pigment extract and its fraction exhibited 82% and 81% cell viability in HDFa cultures, respectively, with no statistically significant differences between treatments, indicating no fibroblast cytotoxicity. In HaCaT cells, viability values of 87.5% and 96% were observed, demonstrating even greater epithelial tolerance. According to ISO 10993-12 criteria [32]—where cell viability above approximately 70–80% is considered non-cytotoxic—these results confirm that both the extract and the pigment fraction exhibit excellent biocompatibility and minimal cytotoxicity toward human skin cell models (Figure 8). Largely, the combination of anti-tyrosinase, anti-inflammatory, antimicrobial, and non-cytotoxic properties highlights a synergistic bioactivity profile that reinforces the cosmetic and dermatological potential of *S. parvulus*-derived pigments.

Additionally, the photostability results revealed that light exposure was the primary factor contributing to pigment degradation, particularly during the first 2 h, when the residual pigment content decreased to approximately 71%. This rapid initial decline, followed by stabilization, suggests that the pigment undergoes photo-oxidative reactions leading to partial chromophore degradation, after which a steady-state equilibrium is established. Similar behavior has been reported for natural pigments, such as anthocyanins and prodigiosin derivatives, in which light exposure induces structural rearrangements or oxidative cleavage of conjugated bonds [35,36]. In contrast, the pigment fraction stored in darkness not only retained its color intensity but also exhibited a slight increase in absorbance (111.68%), likely due to photo-independent tautomerization or solvent-mediated stabilization of the chromophore system, as observed for melanin-like pigments [37]. These findings emphasize the need for light protection during formulation and storage to preserve pigment stability and functional properties.

Thermal stability assays revealed that the pigment-rich fraction remained remarkably stable up to 90 °C, with no significant differences (*p* > 0.05) in residual content compared with room temperature. This behavior contrasts with that of many thermolabile natural colorants, such as betalains and anthocyanins, which typically degrade above 60 °C [38]. The observed resilience may stem from conjugated aromatic or amide-linked structures that prevent oxidative ring cleavage, a feature previously described for *Streptomyces*-derived polyketide pigments [16]. Such thermostability underscores the suitability of this pigment for industrial processes requiring elevated temperatures.

Regarding pH stability, the pigment fraction showed maximal stability between pH 5 and 7, with an unexpected increase in absorbance under acidic conditions (pH 1–3). This response is characteristic of pigments that stabilize via protonation of phenolic or imine groups, enhancing conjugation and color intensity [39]. Conversely, strong alkaline conditions (pH ≥ 11) caused pronounced degradation and a color shift toward pink, likely due to deprotonation and disruption of conjugated chromophore systems. Similar instability under alkaline environments has been reported for other microbial pigments such as violacein and prodigiosin, attributed to cleavage of their pyrrole-based backbones [27,40,41]. In general, these findings indicate that the *S. parvulus* pigment fraction exhibits excellent heat and near-neutral pH stability but remains sensitive to photolytic and alkaline degradation. Such stability characteristics enhance its promise as a natural, biocompatible colorant suitable for cosmetic and food formulations, provided that environmental parameters are carefully controlled during processing and storage.

LC–MS analysis of the pigment-rich fraction from the optimized *Streptomyces parvulus* S145 culture revealed three main 1,4-naphthoquinone-containing polyketides **1**–**3**, such as juglomycin Z, WS-5995B, and naphthopyranomycin, respectively, which together plausibly account for the yellow–orange pigmentation and the fraction’s bioactivity profile. Juglomycin-type 1,4-naphthoquinones are well-documented for their broad antibacterial activity, consistent with historical reports that juglomycin Z and related chemical variants inhibit both Gram-positive and Gram-negative bacteria, as well as yeasts [22]. Moreover, *Streptomyces*-derived juglomycins showed low-MIC activity against both Gram-positive and Gram-negative bacteria, and they can disrupt bacterial cell walls or inhibit essential bacterial enzymes [42]. On the other hand, WS-5995B and its congeners have likewise been linked to antimicrobial and antifungal properties in *Streptomyces* extracts, supporting a plausible role for WS-5995B in the antimicrobial activity observed for the pigment fraction [43]. Beyond direct antibacterial effects, the chemical class represented by these metabolites, i.e., 1,4-naphthoquinones and related aromatic polyketides, has been associated in multiple studies with anti-inflammatory activity (e.g., inhibition of pro-inflammatory signaling and cytokine release) and with tyrosinase inhibition in vitro. These activities have been reported across naphthoquinone derivatives, providing a plausible mechanistic link to the anti-inflammatory properties [44] and structural effects on anti-tyrosinase activity [45], which can also be linked to the observed properties of the crude pigment fraction. Indeed, 1,4-naphthoquinone derivatives are known to modulate oxidative stress and inflammation via several mechanisms, since they may inhibit pro-inflammatory signaling (for example, NF-κB or MAPK pathways) by altering redox balance, inducing antioxidant response elements such as Nrf2, or scavenging ROS [46]. For tyrosinase inhibition, studies have revealed that substituted 1,4-naphthoquinones and other quinone derivatives can bind to the active site of tyrosinase, particularly interacting with its copper ion(s) and key histidine residues, and act as competitive or mixed inhibitors [47].

The detection of these 1,4-naphthoquinone-like polyketides aligns with the expanding body of literature highlighting *Streptomyces* as a prolific producer of bioactive pigments and polyketides with diverse cosmetic-relevant properties [48]. However, it is essential to emphasize that the current annotations are at MS-level confidence (level 2) [49] and that direct attribution of the fraction’s anti-inflammatory and anti-tyrosinase activities to **1**–**3** requires targeted confirmation. Isolation of the individual metabolites, testing of pure compounds in the relevant bioassays (anti-inflammatory cytokine assays, enzymatic tyrosinase inhibition, and MIC/MBC antibacterial testing), and structure–activity analyses (or MS/MS-guided correlation with bioactivity) would be needed to verify which compound(s) drive each activity and to exclude synergistic effects within the pigment mixture.

Overall, this study demonstrated that pigment production in *S. parvulus* can be optimized by strategic adjustments to medium composition and culture conditions, resulting in a 2.5-fold increase in pigment yield and demonstrating multifaceted biological activities with promising cosmetic applications.

## 4. Materials and Methods

### 4.1. Origin of Test S. parvulus Strain S145

The Bioprospecting Research Group at Universidad de La Sabana preserves an in-house strain collection of Actinobacteria isolated from sediments of the Arauca River (Colombia). These isolates exhibit morphological characteristics typical of *Streptomyces* and have been maintained on ISP2 medium (component (g/L): malt extract (10), glucose (4), yeast extract (4), agar (15)) [19,50]. The selected strain (*Streptomyces parvulus* S145), previously identified as a strong extracellular yellow pigment producer [19], was used for all experiments.

#### 4.1.1. Preparation of the Supernatant-Derived Extract

The selected *S. parvulus* S145 strain was thawed and reactivated from cryopreserved stock (ISP2 with 40% glycerol, stored at −70 °C in a Revco high-performance freezer (Thermo Scientific, Waltham, MA, USA) [51]. Spores were streaked on ISP2 agar plates and incubated at 30 °C for 7 days. A 1 cm^2^ agar plug was inoculated into 1-L ISP2 broth and incubated at 30 °C with shaking at 150 rpm for 7 days. Each experiment was performed in biological triplicate [2,13]. Cultures were centrifuged (5000× *g*, 10 min, Sorvall ST16, Thermo Scientific, Waltham, MA, USA) to separate the biomass from the supernatant. Both fractions were freeze-dried (Freezone 2.5 L, Labconco, Kansas City, MO, USA) at 0.220 mbar and −57 °C for 24 h, yielding dry pigmented powders. A portion of the extracellular, lyophilized, pigment-containing powder was used to prepare a 1 mg/mL solution in a 70:30 (*v*/*v*) water:ethanol mixture, filtered through 0.22 µm membranes, and stored at −20 °C until use, designated as the crude hydroalcoholic extract [52]. A second portion of this lyophilized powder was extracted with ethyl acetate to obtain a pigment-retaining fraction composed of three components (vide supra), designated as the pigment-rich fraction. This fraction was concentrated to dryness and stored at −20 °C until use.

#### 4.1.2. Pigment Quantification as Response Variable

Pigment quantification was performed spectrophotometrically on the pigment-rich fraction. A preliminary experiment was first conducted to determine the maximum absorption wavelength of the pigment-rich fraction produced under the reported conditions [19]. The previously obtained pigment-rich fraction was dissolved in ethanol (0.6 mg/mL). The absorbance spectrum of this ethanolic solution was recorded using a spectrophotometer (Thermo Scientific, Waltham, MA, USA) over 300–800 nm with a quartz cuvette (minimum volume of 150 µL) (Sigma-Aldrich, St. Louis, MO, USA). Maximum absorption was observed at 450 nm (Appendix A). A standard curve at 450 nm was subsequently generated using ten 1:2 serial dilutions (1000–1.95 µg/mL) in ethanol. Aliquots of 100 µL were analyzed in triplicate in 96-well plates and read on a microplate reader (Bio-Rad, Hercules, CA, USA). Pigment concentrations (µg/mL) in experimental samples were quantified based on absorbance at 450 nm of ethanolic solutions of dry ethyl acetate fractions obtained from lyophilized fermentation supernatants. Calculations were performed using the linear regression equation derived from the standard curve: A_450nm_ = 0.0005 × [pigment (µg/mL)] + 0.0553 (R^2^ = 0.9865).

### 4.2. Test Factors: Plackett–Burman Experimental Design

To identify nutritional and physical factors with the most significant influence on pigment biosynthesis, five carbon (i.e., soluble starch, glycerol, glucose, lactose, and sucrose) and five nitrogen (ammonium sulfate, potassium nitrate, ammonium chloride, yeast extract–malt extract, and casein) sources were evaluated, based on previous reports optimizing pigment production in *Streptomyces* [13,14,15,18,20,53,54]. The source with the most significant effect on pigment yield was selected for factorial design testing. Two concentrations of each source (0.2% (level 1) and 0.1% (level –1)) were included as study factors in a Plackett–Burman design (PBD), which is a two-level factorial design that enables identification of significant variables affecting production [3,20,51]. Six independent factors were evaluated: pH, temperature, agitation speed, incubation time, carbon concentration, and nitrogen concentration, involving the respective conditions at levels 1 and –1. All experiments were conducted in a 30 mL working volume using 1 mg of biomass (dry weight) per mL, in biological triplicate. The pigment-rich fraction was obtained using the same procedure as in the initial experiment, under the reported conditions (*vide supra*). The twelve experimental runs and their coded factor levels are summarized in Table 3.

As previously mentioned, pigment yield (µg/mL) was used as the response variable, and the data were fitted to a first-order regression model (Equation (3)) [3]:(3)$$Y=\beta_{0}+\;\sum \beta_{i}\chi_{i}$$
where *Y* is pigment production, *β*_0_ is the intercept, *β_i_* are the linear coefficients, and *χ_i_* are the coded factors of the design [3].

### 4.3. Box–Behnken Design: Bioproduction Optimization

Variables with the most potent positive effect on pigment yield were subsequently optimized using a Box–Behnken response surface methodology (RSM). A total of 15 runs were conducted (30 mL volume, 1 mg biomass (dry weight)/mL), testing three levels of each factor: low (−1), medium (0), and high (1). The design matrix is presented in Table 4. All tests were conducted in biological triplicate.

The experimental data were fitted to a quadratic polynomial model (Equation (4)) [3]:(4)$$Y=\beta_{0}+\;\sum \beta_{i}\chi_{i}+\sum \beta_{ii}{\chi^{2}}_{i}+\;\sum \beta_{ij}\chi_{i}\chi_{j}$$
where *Y* is pigment production, *β*_0_ is the intercept, *β_i_* are linear coefficients, *β_ii_* are quadratic coefficients, *β_ij_* are interaction coefficients, and *χ_i_* are coded levels of independent variables [3].

Model validation was performed by repeating the optimized conditions in a 1 L custom-designed, stirred-tank bioreactor (Thermo Scientific, Waltham, MA, USA), using a 100 mL working volume, 100 mg of inoculum (dry biomass) to obtain a ratio of 1 mg of biomass/mL of medium, an aeration rate of 0.5 vvm, a stirring speed of 50 rpm, a pH of 7, and a temperature of 30 °C.

### 4.4. Chemical Characterization of the Pigment-Rich Fraction from S. parvulus

LC–MS analysis of pigment-rich fraction was performed through high-performance liquid chromatography (HPLC) coupled to a diode array detector (DAD) on a Prominence Ultra-Fast Liquid Chromatographic (UFLC) system (Shimadzu, Columbia, MD, USA) and a micrOTOF-Q II mass spectrometer (Bruker, Billerica, MA, USA). Fraction solutions were prepared in absolute ethanol, and 10 µL of each solution was injected into the system. Separations were carried out using a Synergi C18 column (4.6 mm × 150 mm, 3.5 µm) (Phenomenex, Torrance, CA, USA). The mobile phases were 0.1% formic acid in acetonitrile (A) and 0.1% formic acid in water (B), with gradient elution: 0–3 min, 10% A; 3–14 min, 10–40% A; 14–21 min, 40–70% A; 21–25 min, 70–100% A; 25–30 min, 100–10% A. Electrospray ionization interface (ESI) was operated in a negative ion mode (scan 100–1200 *m*/*z*), desolvation line temperature at 250 °C, nitrogen as nebulizer gas at 1.5 L/min, drying gas at 8 L/min, quadrupole energy at 7.0 eV, and collision energy 14 eV. ATR-FTIR spectra of freeze-dried extracts were obtained using a Cary 630 spectrometer (Agilent, Santa Clara, CA, USA) over 400–4000 cm^−1^, with 24 scans and a resolution of 4 cm^−1^ [2,55]. Feature annotations at level 2 [49] were performed after a detailed scrutiny of the ultraviolet-visible (UV-Vis), FTIR, and high-resolution mass spectral (exact mass of quasi-molecular and fragment ions) data for each signal, in comparison to the information compiled in the StreptomeDB 4.0 database [56].

### 4.5. 16 S rRNA Gene Sequencing

DNA was extracted using the GF-1 Bacterial DNA extraction kit (Vivantis, Subang Jaya, Malaysia). The 16S rRNA gene was amplified by PCR using primers 27F and 1492R [57,58].

### 4.6. Tyrosinase Inhibition Assay

Tyrosinase inhibition was determined as described previously [8,10], with modifications. Reactions were carried out in 96-well plates using mushroom tyrosinase (200 U/mL, 100 µL), L-DOPA (2.5 mM, 100 µL), and pigment-rich fraction (50 µL) at 30, 300, and 1000 µg/mL. Absorbance was measured at 490 nm after 5 min. Kojic acid (50–1000 µg/mL, 50 µL) was used as a positive control, and 3% DMSO (50 µL) as a negative control. Each condition was tested in quadruplicate. % inhibition was then calculated (Equation (5)).(5)$$\%\;Inhibition=\left\{\left[(A-B)-(C-D)]/(A-B\right)\right\}$$
*A*: Absorbance of the enzyme, substrate, and DMSO solution.*B*: Absorbance of the substrate and DMSO solution.*C*: Absorbance of the enzyme, substrate, and extract solution.*D*: Absorbance of the substrate and extract solution.

### 4.7. Anti-Inflammatory Assay

The anti-inflammatory activity was evaluated using previously described methodologies [59]. The THP-1 cell line (ATCC^®^ TIB-202™) was seeded at 1 × 10^6^ cells per well in 24-well plates and cultured in RPMI medium supplemented with 10% fetal bovine serum (FBS) (Invitrogen, Waltham, MA, USA) for 24 h. Monocytic THP-1 cells were differentiated into macrophages by incubation with 150 nM phorbol 12-myristate 13-acetate (PMA) for 48 h, followed by 24 h in RPMI medium [59].

Macrophages adhered to the bottom of the wells were washed three times with 600 µL PBS (0.1 M, pH 6.4) and incubated in FBS-free RPMI medium with 50 ng/mL lipopolysaccharide (LPS) for 4 h to induce an inflammatory response [60,61,62]. LPS-stimulated THP-1 macrophages were then treated for 24 h with either RPMI medium alone, the crude hydroalcoholic extract, or the ethyl acetate fraction (pigment-rich fraction) at 300 µg/mL. At the end of the incubation, supernatants were collected from monocytic THP-1 cells, differentiated macrophages, LPS-stimulated macrophages, and LPS-stimulated macrophages treated with extract or fraction. Ibuprofen (50 µg/mL, Sigma Chemical Co., St. Louis, MO, USA), added for 24 h after LPS stimulation, was used as a positive control. TNF-α and IL-10 levels in the supernatants were quantified using commercial ELISA kits according to the manufacturer’s instructions (Human IL-10 ELISA Kit (ab185986) and Human TNF-αELISA kit (ab181421) (Abcam Inc., Cambridge, MA, USA)). Absorbance was measured at 450 nm using a microplate reader [59], and cytokine concentrations were calculated from standard curves. All experiments were performed in biological triplicate.

### 4.8. Anti-Acne Activity of the Fractionated Extract

The anti-acne activity of the pigment-rich fraction was evaluated against S. epidermidis (ATCC 12228) using the disk diffusion method, in accordance with the guidelines of the European Committee on Antimicrobial Susceptibility Testing (EUCAST). Standard antibiotic disks (vancomycin 5 µg) and Trypticase Soy Agar (TSA) plates were employed. Plates were surface-inoculated with 100 µL of an *S. epidermidis* suspension equivalent to half a McFarland standard (No. 1). The inoculum was spread evenly over the plate surface in three directions, with a final sweep around the edge. After drying, the disks were seeded with 30 µL of the *S. parvulus* fraction (10 mg/mL), vancomycin (5 µg, Liofilchem) as a positive control, and 30 µL of an ethanol–water mixture (70:30) as a negative control [9,63]. Disks were gently pressed onto the agar, the plates were inverted, and the plates were incubated at 37 °C for 24 h. Inhibition zones were measured in millimeters. Additionally, S. epidermidis growth was monitored using a Bioscreen C analyzer (Growth Curve USA^®^, Piscataway, NJ, USA) under aerobic conditions. Wells of a HoneyComb plate were inoculated with 100 µL of bacterial suspension (1 × 10^6^ CFU/mL) and treated with different concentrations of the fractionated extract (300, 30, and 3 µg/mL). The vehicle (ethanol–water 70:30) served as a negative control, and vancomycin (40, 4, and 0.4 mg/mL) as the positive control. Plates were incubated at 37 °C, and growth was monitored at 30 min intervals for 20 h by measuring the optical density at 600 nm [14].

### 4.9. Cytotoxic Assay

The cytotoxic activity of the pigmented crude extract and its corresponding fraction was evaluated against human dermal fibroblasts (HDFa, passages 3–8) and human keratinocytes (HaCaT, passages 4–9) using the standard MTT assay. Both cell lines were maintained in Dulbecco’s Modified Eagle’s Medium (DMEM; Gibco, Waltham, MA, USA) supplemented with 10% fetal bovine serum (FBS; Gibco, Waltham, MA, USA), 100 U/mL penicillin, and 100 µg/mL streptomycin, and incubated at 37 °C in a humidified atmosphere containing 5% CO_2_. Upon reaching approximately 80–90% confluence in 75 cm^2^ culture flasks, cells were detached using 0.25% trypsin–EDTA, counted with a hemocytometer, and seeded at a density of 1 × 10^5^ cells per well in 96-well microplates. After a 24 h attachment period, both cell lines were treated with the crude extract or pigment-rich fraction at a final concentration of 300 µg/mL. Treatments were performed in quintuplicate (*n* = 5) and incubated for 24 h under standard culture conditions. Following exposure, the culture medium was replaced with 100 µL of MTT solution (5 mg/mL in phosphate-buffered saline; Sigma-Aldrich, St. Louis, MO, USA) and incubated for 4 h at 37 °C in the dark. After incubation, the supernatant was discarded, and the resulting formazan crystals were solubilized by adding 100 µL of dimethyl sulfoxide (DMSO; Merck, Germany) to each well. Absorbance was measured at 570 nm using a microplate reader (Bio-Rad, Hercules, CA, USA). DMSO solutions at 5% and 10% served as positive cytotoxicity controls [50], while untreated cells were used as the negative control. Cell viability (%) was calculated relative to untreated controls using the following Equation (6):(6)$$\mathrm{Cell}\;\mathrm{viability}\;(\%)=\frac{A_{\mathrm{sample}}}{A_{\mathrm{control}}}\times 100$$

### 4.10. Stability Assessment of the Pigment-Rich Fraction

The concentration of the pigment in each sample was quantified by measuring optical density (OD) at 450 nm using a UV–visible spectrophotometer (Bio-Rad, Hercules, CA, USA) [64]. All pigment solutions were freshly prepared at a standardized concentration of 1 mg/mL in distilled water. The pigment residual ratio, indicating the percentage of pigment remaining after each treatment, was calculated by comparing the OD values before and after exposure according to the formula described by Zhu et al. [65].

To evaluate photostability, aliquots of the pigment solution were exposed to indoor incandescent light at ambient temperature for 2–12 h, while parallel samples were stored in complete darkness as controls [65]. For thermal stability, samples were incubated in a temperature-controlled water bath (Memmert, Schwabach, Germany) at 0, 4, 20, 37, 60, and 90 °C for 2 h, then cooled to room temperature before OD measurement. To determine pH stability, aqueous solutions of the yellow pigment fraction were adjusted to different pH values (1.0, 3.0, 5.0, 7.0, 9.0, 11.0, and 13.0) using 0.1 M HCl or 0.1 M NaOH, and maintained in a light-protected environment at room temperature for 2 h before absorbance determination [65]. All experiments were replicated using four technical replicates (*n* = 4), and results were expressed as mean ± standard deviation (SD).

### 4.11. Statistical Analysis

The bioactivity data were first assessed for normality using the Shapiro–Wilk test (*p* > 0.5), and then analyzed using one-way analysis of variance (ANOVA). When significant differences were detected, Tukey’s HSD post hoc test was applied for multiple comparisons (*n* = 3). Statistical analyses were performed using GraphPad Prism 9 (GraphPad Software Inc., Boston, MA, USA). Results are reported as mean ± standard deviation (SD), with significance set at *p* < 0.05. For model evaluation, the adequacy of the polynomial equation was assessed using the correlation coefficient (R), and the F test further confirmed its statistical significance in Design-Expert v13 (Stat-Ease Inc., Minneapolis, MN, USA).

## 5. Conclusions

This study demonstrates that *Streptomyces parvulus*, isolated from sediments of the Arauca River in Colombia, is a promising microbial source of bioactive pigments with potential cosmeceutical applications. Statistical optimization using a Box–Behnken design identified 30 °C, 50 rpm, and 7 days of incubation as the most favorable conditions for maximizing extracellular pigment yield. The predictive model showed strong reliability, with a validation error of only 1.72%, underscoring the robustness of the optimization strategy. The pigment-rich fraction exhibited excellent biocompatibility with human fibroblasts and keratinocytes, as well as remarkable stability under heat and at near-neutral pH. Significant degradation was observed only upon prolonged light exposure or under strongly alkaline environments. Samples stored in darkness maintained—or even enhanced—their color intensity, confirming the pigment’s robustness under controlled conditions. These findings indicate that the pigment fraction is thermally stable, pH-resilient, and biocompatible, highlighting its promising potential as a natural colorant for food, cosmetic, and biomedical applications.

Chemical characterization of the pigment-rich fraction by LC–MS and FTIR suggested the presence of secondary metabolites, including juglomycin Z, WS-5995B, and naphthopyranomycin. These three 1,4-naphthoquinone-containing compounds are plausibly responsible for the observed biological properties. Thus, the fraction exhibited tyrosinase inhibitory activity, anti-inflammatory effects in LPS-stimulated macrophages, and anti-acne activity against *Staphylococcus epidermidis*, highlighting its multifunctional bioactivity profile. Considered together, these findings establish *S. parvulus* S145-derived pigments as attractive candidates for the development of natural, multifunctional colorants for cosmeceutical formulations. Beyond replacing synthetic dyes, such pigments provide added biological value, aligning with current consumer demand for sustainable, bio-based, and health-promoting ingredients. Further work should focus on scale-up fermentation strategies, purification of individual active compounds, and in vivo efficacy and safety validation to fully realize their industrial potential.

## Figures and Tables

**Figure 1 ijms-26-10762-f001:**
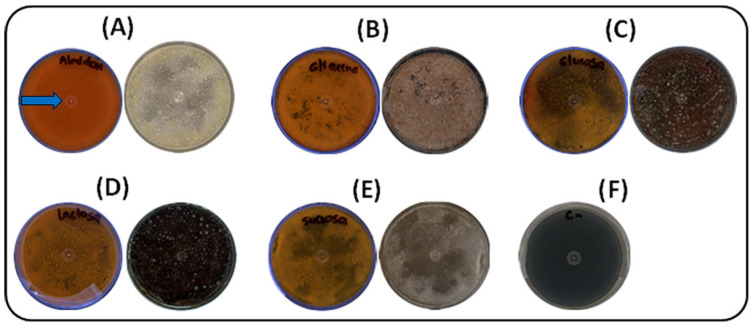
Effect of carbon sources on growth and pigment production by strain S145 at 30 °C after seven days of incubation. For each source, the lower (left) and upper (right) parts of the Petri dish are shown. (**A**) Soluble starch, (**B**) Glycerol, (**C**) Glucose, (**D**) Lactose, (**E**) Sucrose, (**F**) Negative control (ISP2, without inoculum). Plate A shows the yellow pigment selected for further analysis, as indicated by the blue arrow.

**Figure 2 ijms-26-10762-f002:**
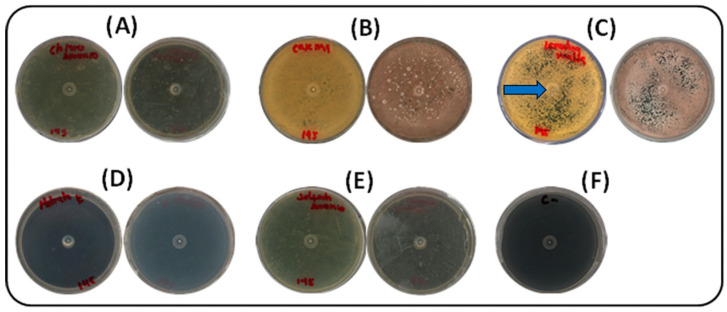
Effect of nitrogen sources on growth and pigment production by strain S145 at 30 °C after seven days of incubation. For each source, the lower (left) and upper (right) parts of the Petri dish are shown. (**A**) Ammonium chloride, (**B**) Casein, (**C**) Yeast/malt extract, (**D**) Potassium nitrate, (**E**) Ammonium sulfate, (**F**) Negative control (ISP2, without inoculum). Plate C shows the yellow pigment selected for further analysis, as indicated by the blue arrow.

**Figure 3 ijms-26-10762-f003:**
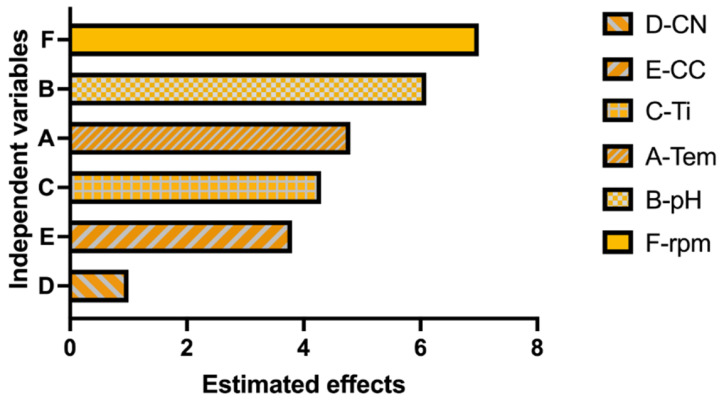
Pareto chart showing the standardized effects of the evaluated factors on pigment production according to the Plackett–Burman design. Factors: A (Temperature), B (pH), C (incubation time), E (Carbon concentration), D (Nitrogen concentration), F (agitation). The vertical line indicates the statistical significance threshold (*p* < 0.05). Data were derived from biological triplicates (*n* = 3).

**Figure 4 ijms-26-10762-f004:**
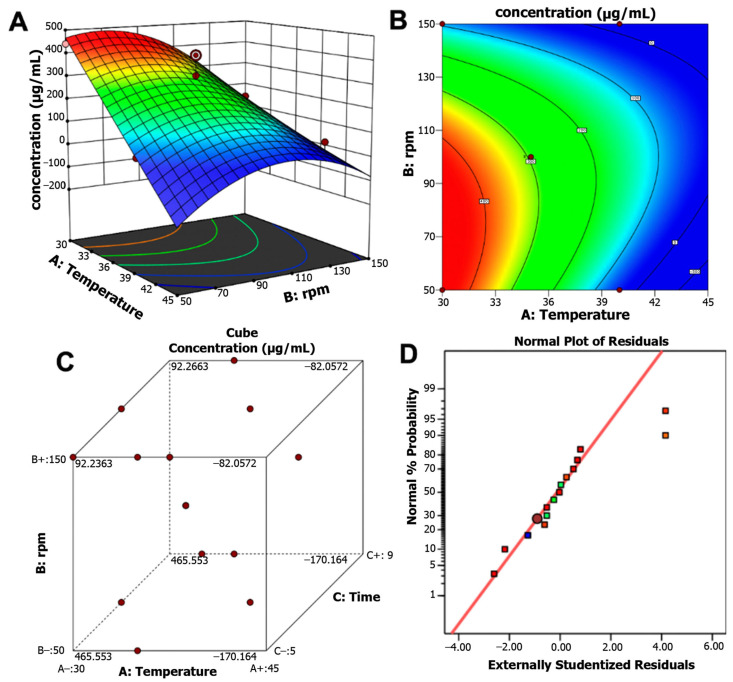
(**A**) Three-dimensional response surface plot. (**B**) Contour map for pigment production of *Streptomyces* strain S145 showing the interactive effects of temperature, stirring speed, and fixing the incubation time variable. (**C**) Pigment production concentration cube at different design points. (**D**) Normal plot of the residuals, indicating the quality of the optimized model.

**Figure 5 ijms-26-10762-f005:**
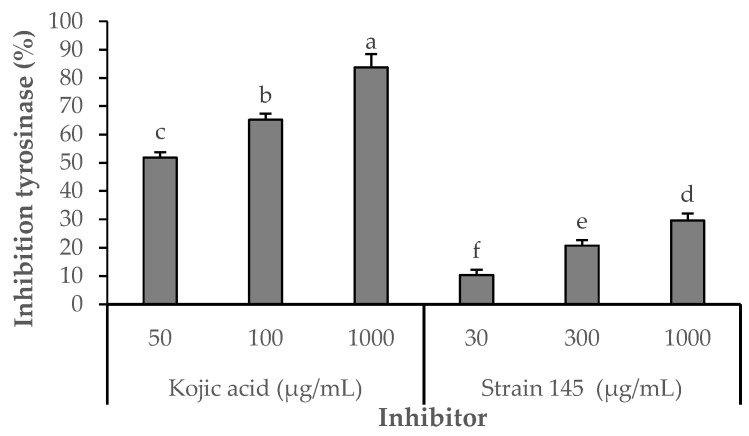
Effects of Kojic acid or fraction S145 on tyrosinase diphenolase activity. Enzyme activity was tested in the presence of L-DOPA as substrate. After incubation, the amount of dopachrome produced was determined spectrophotometrically at 490 nm. Results are presented as means ± SD of four experiments. Different letters indicate statistically significant differences (*p* < 0.05) as determined by Student’s *t*-test. Strain S145 corresponds to the derived fraction of extracellular pigment produced by *Streptomyces parvulus* strain. Kojic acid was used as the positive control. The negative control (DMSO 3%) showed 0% inhibition.

**Figure 6 ijms-26-10762-f006:**
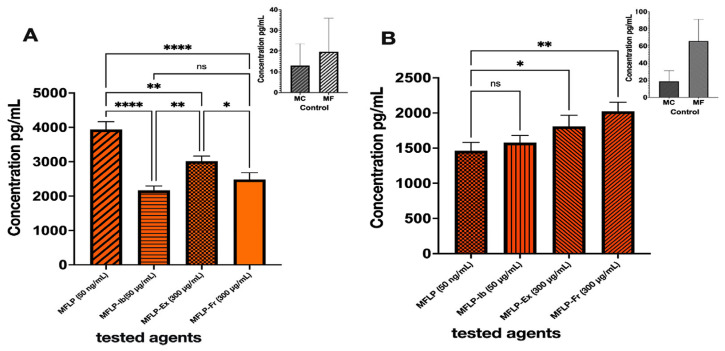
Effect of aqueous/ethanolic colored extract and pigment fraction of *Streptomyces* S145 on cytokine production. (**A**). TNF-α; (**B**). IL-10. Bars represent mean ± SD (*n* = 3). Different bar patterns indicate distinct treatments. Asterisks denote significant differences between treatments and the control according to one-way ANOVA followed by Tukey’s test (* *p* < 0.05; ** *p* < 0.01; **** *p* < 0.0001; ns = not significant.

**Figure 7 ijms-26-10762-f007:**
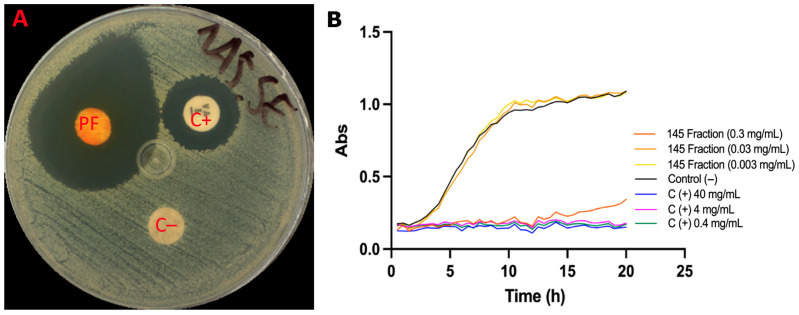
Effect of the fractionated pigment from *Streptomyces parvulus* against *S. epidermidis*. (**A**). Disk diffusion assay: 30 µL of the water–ethanol vehicle (70:30) was applied as a negative control (C−); 30 µL of the pigmented fraction (PF) (10 mg/mL) was tested; and a disc with vancomycin (5 µg, Liofilchem^®^, Piane Vomano, Italy) was used as the positive control (C+). Inhibition halos were measured in millimeters. (**B**). Growth kinetics under aerobic conditions: Wells were inoculated with 100 µL of *S. epidermidis* (1 × 10^6^ CFU/mL) and treated with the fractionated extract at 300, 30, or 3 mg/mL. The vehicle (70:30 water–ethanol) served as the C−, and vancomycin at 40, 4, and 0.4 mg/mL was used as C+. Cultures were incubated at 37 °C for 20 h, with OD_600_ readings taken every 30 min.

**Figure 8 ijms-26-10762-f008:**
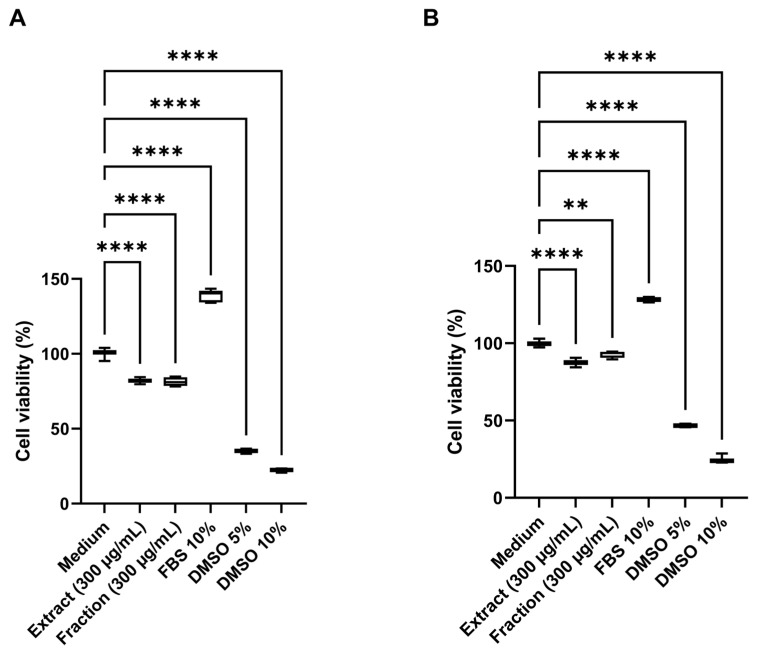
Cytotoxic activity of the pigmented extract and fraction at 300 µg/mL. (**A**) Cell viability (%) of human dermal fibroblasts (HDFa) after 24 h of treatment. (**B**) Cell viability (%) of human keratinocytes (HaCaT) after 24 h of treatment. Medium supplemented with 10% fetal bovine serum (FBS) was used as a cell growth control (negative cytotoxicity control), while dimethyl sulfoxide (DMSO) at 5% and 10% served as positive cytotoxicity controls. Data are expressed as mean values ± standard deviation (*n* = 5). **** and ** indicate significant differences at *p* < 0.0001 and *p* < 0.001, respectively, according to analysis of variance (ANOVA) followed by Dunnett’s multiple comparison test.

**Figure 9 ijms-26-10762-f009:**
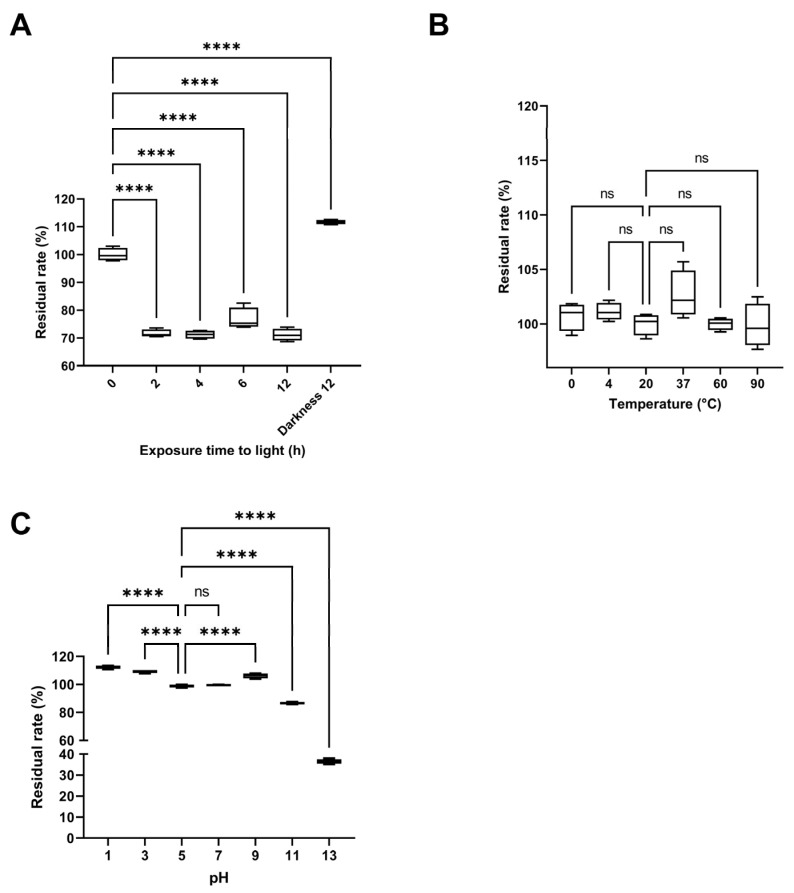
Stability assays of the pigment-rich fraction from *Streptomyces parvulus*. (**A**). Stability of yellow pigment solutions after exposure to artificial light for 0, 2, 4, 6, and 12 h, and under darkness for 12 h (Darkness 12). (**B**). Thermal stability of yellow pigment solutions at different temperatures (0, 4, 20, 37, 60, and 90 °C) after 2 h of treatment. (**C**). Stability of yellow pigment solutions at different pH values (1, 3, 5, 7, 9, 11, and 13) after 2 h of treatment. All treatments were performed with 5 mL of a 1 mg/mL pigment-rich fraction in water. Data are expressed as mean ± standard deviation (*n* = 4). “ns” indicates no significant difference, and **** indicates a significant difference (*p* < 0.0001), according to analysis of variance (ANOVA) followed by Dunnett’s multiple comparisons test.

**Table 1 ijms-26-10762-t001:** Factorial regression of the response variable: Pigment concentration (µg/mL) vs. Factors: Temperature, pH, N source (%), C source (%), agitation (rpm), and incubation time (Ti).

Source	DF	Adjust SC	Adjust MC	F-Value	*p*-Value
Model	6	10,314.6	1719.1	24.12	0.002
Lineal	6	10,314.6	1719.1	24.12	0.002
Temp	1	1587.0	1587.00	22.27	0.005
pH	1	2760.3	2760.33	38.74	0.002
Ti	1	1382.5	1382.45	19.4	0.007
CN	1	62.20	62.2	0.87	0.393
CC	1	1077.7	1077.69	15.12	0.012
rpm	1	3444.92	3444.92	48.34	0.001
Error	4	71.26	71.26		
Total	11				
		S	R^2^	R^2^ (Adjust)	R^2^ (Pred)
		8.44148	96.66%	92.65%	80.77%

**Table 2 ijms-26-10762-t002:** Regression statistics and analysis of variance were used to analyze the Box–Behnken results and optimize pigment production by the *Streptomyces* strain S145. * Significant values, df: Degree of freedom, F: Fisher’s function, *p*: Level of significance, C.V: Coefficient of variation.

Source	Sum of Squares	Df	Mean Square	F-Value	*p*-Value
Model	3.117 × 10^5^	1	77,920.09	22.66	<0.0001 *
A—Temperature	1.367 × 10^5^	1	1.367 × 10^5^	39.76	<0.0001 *
B—rpm	27,109.21	1	27,109.21	7.88	0.0185 *
AB	27,888.84	1	27,888.84	8.11	0.0173 *
B2	78,613.75	1	78,613.75	22.87	0.0007 *
Residual	34,381.57	10	3438.16		
Lack of Fit	17,814.71	8	2226.84	0.2688	0.9279
Pure Error	16,566.86	2	8283.43		
Cor Total	3.461 × 10^5^	14			
St Dev.	58.64	R^2^	0.9006		
Mean	191.81	fitted R^2^	0.8609		
C.V %	30.57	Predicted R^2^	0.7914		
PRESS	72,193.05	Adeq precision	14.4596		

**Table 3 ijms-26-10762-t003:** Plackett–Burman experimental design for evaluating independent variables with coded values and pigment production (µg/mL). T (temperature); pH; Ti (incubation time); rpm (stirring speed); C.N (nitrogen concentration); C.C (carbon concentration).

Run		T (°C)	pH	Ti (days)	C. N (%)	C. C (%)	rpm	Pigment (µg/mL)
1		1	−1	1	−1	−1	−1	85.33
2		−1	1	1	−1	1	−1	13.33
3		1	1	−1	1	−1	−1	43.40
4		1	−1	1	−1	1	1	7.40
5		−1	1	1	1	1	−1	62.00
6		1	−1	1	1	−1	1	36.07
7		1	1	−1	−1	1	1	12.00
8		1	−1	−1	1	1	−1	6.73
9		−1	−1	−1	1	1	1	60.07
10		−1	−1	1	1	−1	1	70.73
11		−1	1	−1	−1	−1	1	78.73
12		−1	−1	−1	−1	−1	−1	0.07
Levels	1	35	8	7	0.2	0.2	200	
	−1	25	6	3	0.1	0.1	100	

**Table 4 ijms-26-10762-t004:** Box–Behnken response surface design represents the extracellular pigment production (µg/mL) of *S. parvulus* as a function of the main effects.

Run		T (°C)	rpm	Ti (days)	Pigment (µg/mL)
1		−1	0	1	445.00
2		−1	1	0	142.00
3		1	−1	0	42.73
4		1	0	−1	76.40
5		0	0	0	210.40
6		−1	−1	0	440.06
7		0	−1	−1	247.40
8		1	0	1	205.40
9		0	0	0	304.10
10		0	1	1	8.70
11		1	1	0	15.40
12		0	−1	1	284.40
13		0	1	−1	7.40
14		0	0	0	392.40
15		−1	0	−1	55.40
Levels	1	40	150	9	
	0	35	100	7	
	−1	30	50	5	

## Data Availability

The original contributions presented in this study are included in the article/Appendix A. Further inquiries can be directed to the corresponding author.

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
