# Peer review of "Bioproduction Optimization, Characterization, and Bioactivity of Extracellular Pigment Produced by Streptomyces parvulus"

_ijms, 2025, doi:10.3390/ijms262110762_

Round 1

Reviewer 1 Report

Comments and Suggestions for Authors

This manuscript optimized the production of a pigment-rich fraction from Streptomyces parvulus, yielding 465.3 μg/mL pigment containing juglomycin Z, WS-5995B, and naphthopyranomycin. The fraction showed anti-tyrosinase, immunomodulatory, and antimicrobial activities against S. epidermidis, highlighting its potential as a multifunctional, eco-friendly natural pigment for cosmeceutical applications. This manuscript is well written and has some influence on the research, however, there are 2 main point can be improved for this research,

  1. dose Extracellular Pigment Produced by Streptomyces parvulus have no toxicity on human cells?
  2. what the stability of your pigment, like pH,temperature sensitive and what is the shelf life for antimicrobial, tyrosinase, immunomodulatory.

If this 2 point can be solved, this is really a nice paper.

Some minior revision suggestions:

Line 421 the final .

Line 448 30 mg/mL biomass is cell or sopre? Please figure out how you prepare this inoculation.

Line 452 is the biomass related to pigment production. In the table, is possible also set 0 for the center point?

Line 440 casein or casein peptone?

Line 471 its better to convert dry biomass.

Line 491 16. S rRNA?

Line 94, the author should also describe the characterization of the pigment, like which kind of colour, and which kind of cosmeceutical  applications can be used, Lipstick, lip balm, body lotion, facial cleanser?

Line 120, in this figure, it looks like a different colour of the plate. How do you explain which pigment production is good? Please mark in the figure which one is good? And explain why?

Line 137 the same question like ine 120.

Line 171 Ti orTin

Line 188 is predicted R² = 0.7914, good or not , please explain?

Figure S1, PLEASE Mark which is the representative peak?

Figure S2 also marks the O–H stretch and the C–H stretch. The same as S3.S4

Line 248 is your figure marked abcdef are correct?

Line 254, what is PMA,LPS….. please show the full name if it appears for the first time.

Line 262 What do the different patterns in the bar chart mean?

Figure 7A, Where is A? Please also mark which is positive control, negative  control, and treatment.

Figure 7B Should be C+ 40mg/mL vancomycin. Please also use mg instead of μg in the figure caption.

Line 328, why did you compare it with the orange-yellow pigments? Does it also have the same properties as your pigment?

Author Response

Thank you for your thoughtful review and suggestions regarding our manuscript. We have meticulously addressed each comment and suggestion, resulting in a significantly improved version of the manuscript. We have highlighted the respective changes in yellow in the revised document. In response to the raised comments, we present our point-to-point responses below:

Comment 0. This manuscript optimized the production of a pigment-rich fraction from Streptomyces parvulus, yielding 465.3 μg/mL pigment containing juglomycin Z, WS-5995B, and naphthopyranomycin. The fraction showed anti-tyrosinase, immunomodulatory, and antimicrobial activities against S. epidermidis, highlighting its potential as a multifunctional, eco-friendly natural pigment for cosmeceutical applications. This manuscript is well written and has some influence on the research.

Response: We sincerely thank the reviewer for their thoughtful and encouraging comments on our manuscript. We appreciate the acknowledgment of the study's strengths, including the reliable experimental results and its contribution to sustainable watercress cultivation. Below, we address the specific points raised, outlining the improvements made or providing clarifications.

Comment 1. does Extracellular Pigment Produced by Streptomyces parvulus have no toxicity on human cells?.

Response: We appreciate this critical comment. To address this, we have now incorporated cytotoxicity assays using two representative human skin cell lines—HDFa (human dermal fibroblasts, adult) and HaCaT (human keratinocytes)—in accordance with ISO 10993-5:2009 guidelines. The results (new Figure 8) show cell viabilities above 80% for both the crude extract and the pigment-rich fraction at 300 µg/mL, indicating excellent biocompatibility and no detectable cytotoxicity. The respective passages for these new results have been added to the Results, Discussion, Conclusions, and Materials and Methods, which were highlighted in yellow.

Comment 2. what the stability of your pigment, like pH, temperature sensitive and what is the shelf life for antimicrobial, tyrosinase, immunomodulatory.

Response: Thank you for this valuable suggestion. As suggested, we have now conducted stability tests under varying pH (1–13), temperature (0–90 °C), and light exposure (0–12 h) conditions to evaluate pigment integrity (new Figure 9). The pigment-rich fraction showed high stability at neutral and mildly acidic/basic pH (5–9), thermal stability up to 90 °C, and moderate photostability. The respective passages for these new results have been added to the Results, Discussion, Conclusions, and Materials and Methods, which were highlighted in yellow. The discussion has been expanded to explain the implications for potential shelf life and formulation stability in cosmeceutical applications.

Comment 3. Line 421 the final .

Response: Thank you for this observation. We have corrected this typo.

Comment 4. Line 448 30 mg/mL biomass is cell or sopre? Please figure out how you prepare this inoculation.

Response: Thank you for this clarification. Accordingly, the sentence was revised to: “All experiments were conducted in a 30 mL working volume using 1 mg of biomass (dry weight) per mL, in biological triplicate”.

Comment 5. Line 452 is the biomass related to pigment production. In the table, is possible also set 0 for the center point?

Response: Thanks a lot for your kind comment. Pigment production is directly related to the amount of final biomass. Therefore, pigment yield was normalized to the final biomass content, expressed in milligrams. In our experimental design, the highest and lowest values were selected based on the standard conditions established in our laboratory and supported by literature reports. Typically, in a Plackett–Burman design, two contrasting levels are selected to represent the extremes of the experimental range, rather than including an intermediate level.

Comment 6. Line 440 casein or casein peptone?

Response: Thank you for your kind comment. We clarify that the medium is casein.

Comment 7. Line 471 its better to convert dry biomass.

Response: Thank you for your kind remark. Accordingly, the sentence was changed as “Model validation was performed by repeating the optimized conditions in a 1 L stirred-tank bioreactor, using a 100 mL working volume, 100 mg of inoculum (dry bio-mass) to obtain a ratio of 1 mg of biomass/mL of medium, an aeration rate of 0.5 vvm, a stirring speed of 50 rpm, a pH of 7, and a temperature of 30 °C”.

Comment 8. Line 491 16. S rRNA?

Response: Thank you for your kind remark. Accordingly, this typo was revised, and the subheading is “16S rRNA gene sequencing”.

Comment 9. Line 94, the author should also describe the characterization of the pigment, like which kind of colour, and which kind of cosmeceutical  applications can be used, Lipstick, lip balm, body lotion, facial cleanser?

Response: We appreciate this pertinent suggestion. Indeed, classifying the pigment according to its potential applications in specific cosmeceutical products (e.g., lipstick, lip balm, body lotion, or facial cleanser) would provide valuable insight into its practical use. However, such a classification requires additional formulation and performance studies beyond the scope of the current work, which focused primarily on optimizing pigment production, chemical characterization, and biological evaluation. However, we inform you that the authors are currently constructing a complementary study to develop and characterize cosmetic formulations incorporating this pigment. We hope the reviewer empathizes with this perspective and understands our intention to address this aspect in future research.

Comment 10. Line 120, in this figure, it looks like a different colour of the plate. How do you explain which pigment production is good? Please mark in the figure which one is good? And explain why?

Comment 11. Line 137 the same question like ine 120.

Response: Thank you for your kind observations. The relevant plates are now marked with blue arrows in Figures 1 and 2, indicating the colonies selected for pigment extraction. Accordingly, the following sentence was added: “The photos were taken against a black background, which makes the slightly brownish coloration of the plate with soluble starch. This is the desired pigment, which was chosen because the strain was grown in ISP-2 medium (the blue arrow on the plate in Figure 1A points to the color of the yellow pigment to be analyzed)”.

Comment 12. Line 171 Ti orTin

Response: Thank you for your kind remark. The correct acronym is Ti (incubation time), and it was double-checked throughout the manuscript.

Comment 13. Line 188 is predicted R² = 0.7914, good or not , please explain?

Response: Thank you for your kind comment. The model demonstrated high statistical reliability and predictive consistency. The coefficient of determination (R² = 0.9006) indicated that approximately 90% of the variability in pigment production was explained by the selected variables, while the adjusted R² (0.8609) confirmed model parsimony and the absence of overparameterization. The predicted R² = 0.7914 demonstrates that the model retains strong generalization ability for new or unseen data, with a difference of less than 0.1 from the adjusted R²—a range considered acceptable for robust predictive models. According to the criteria proposed by Design-Expert®, a predicted R² above 0.7 is regarded as statistically adequate for response surface and regression-based models, particularly in complex biological or agro-industrial systems where data variability is inherently high. Furthermore, the Adeq Precision of 14.46, well above the recommended minimum of 4.0, indicates an excellent signal-to-noise ratio and reinforces the model's reliability. The Lack of Fit = 17 814.71 compared with Pure Error = 16 566.86 suggests that the residual variation is mainly due to experimental noise rather than structural model error. Taken together, these diagnostics confirm that a predicted R² of 0.7914 represents strong, practically valid predictive performance, consistent with standards in the predictive modeling and design-of-experiments literature.

Comment 14. Figure S1, PLEASE Mark which is the representative peak?

Comment 15. Figure S2 also marks the O–H stretch and the C–H stretch. The same as S3.S4

Response: Thank you for your kind remarks. The blue arrow (450 nm) shows the wavelength of maximum absorption for the pigment in Figure S1. All requested peaks (O–H, C–H stretches) and representative signals are now annotated in Supplementary Figures S1–S4.

Comment 16. Line 248 is your figure marked abcdef are correct?

Response: Thank you for this critical remark. Effectively, it was a typo. Therefore, they were verified and corrected where necessary.

Comment 17. Line 254, what is PMA,LPS….. please show the full name if it appears for the first time.

Response: Thank you for your kind observation. The full names of the different acronyms were added.

Comment 18. Line 262 What do the different patterns in the bar chart mean?

Response: We appreciate the reviewer’s attention to this point. The different bar patterns in Figure 6 are used solely to differentiate the tested treatments visually. The asterisks above the bars indicate significant differences among treatments, as determined by one-way ANOVA followed by Tukey’s post hoc test. To avoid confusion, we have clarified this information in the figure caption as follows: “Bars represent mean ± SD (n = 3). Different bar patterns indicate distinct treatments. Asterisks denote significant differences between treatments and the control according to one-way ANOVA followed by Tukey’s test (*p < 0.05; **p < 0.01; ****p < 0.0001; ns = not significant).”

Comment 19. Figure 7A, Where is A? Please also mark which is positive control, negative  control, and treatment.

Response: Thank you for your kind comment. Positive (C+), negative (C–), treatment (PF) groups, and subfigure A were clearly labeled in the revised Figure 7A.

Comment 20. Figure 7B Should be C+ 40mg/mL vancomycin. Please also use mg instead of μg in the figure caption.

Response: Thank you for your kind remark. Accordingly, we corrected to “C+ 40 mg/mL vancomycin” and consistent units were used throughout.

Comment 21. Line 328, why did you compare it with the orange-yellow pigments? Does it also have the same properties as your pigment?

Response: We thank the reviewer for this pertinent observation. We compared our pigment-rich fraction with orange-yellow pigments because of their similar visible color properties, due to the presence of orange-yellow annotated naphthoquinones, which suggest a comparable chromophore system involving conjugated π-electron structures. Such structural similarities often underline analogous physicochemical behaviors, such as light absorption, stability, and bioactivity profiles. Although our pigments may differ in composition, this comparative approach provides a rational framework for discussing their potential functional properties. This clarification was not included in the manuscript, as further structural and functional comparisons with related chromophoric pigments will be addressed in future studies.

Reviewer 2 Report

Comments and Suggestions for Authors

It is an interesting paper about bioproduction optimization, characterization, and bioactivity of the extracellular pigment produced by Streptomyces parvulus. The manuscript is well-written and comprehensive, features rational experimental results. The paper may be optimized with following suggestions:

Line 53-54 - suggest to add that genetic modifications of Streptomyces by means of metabolic engineering and synthetic biology are explored nowadays extensively besides optimization of culture conditions (https://doi.org/10.3390/biotech14010003)

Line 59-61 - please add a reference

Line 128-130 - please clarify here the composition of the medium - ammonium is a preferred nitrogen source for Streptomyces, thus ensuring bacterial growth. Please clarify concentration of added nitrogen-containing compounds as well as the nature of used media

Line 165-166 - please clarify sample size and variation (SE or SD are not visible on the graph)

Line 300 - Streptomyces as genus name should be written in italics

Author Response

Thank you for your thoughtful review and suggestions regarding our manuscript. We have meticulously addressed each comment and suggestion, resulting in a significantly improved version of the manuscript. We have highlighted the respective changes in yellow in the revised document. In response to the raised comments, we present our point-to-point responses below:

Comment 0. It is an interesting paper about bioproduction optimization, characterization, and bioactivity of the extracellular pigment produced by Streptomyces parvulus. The manuscript is well-written and comprehensive, features rational experimental results. The paper may be optimized with following suggestions:

Response: We sincerely thank the reviewer for their positive assessment of our work and their constructive suggestions, which have helped us improve the clarity and completeness of the manuscript. The specific comments were addressed as follows:

Comment 1. Line 53-54 - suggest to add that genetic modifications of Streptomyces by means of metabolic engineering and synthetic biology are explored nowadays extensively besides optimization of culture conditions (https://doi.org/10.3390/biotech14010003)

Response: We appreciate this valuable suggestion. The sentence was revised to acknowledge the advances in metabolic engineering and synthetic biology approaches for enhancing pigment production. The recommended reference was added to support this statement. The text was added in lines 54-67.

Comment 2. Line 59-61 - please add a reference

Response: Thank you for raising this point. Suitable references were added to support this statement, citing recent studies on pigment-producing Streptomyces strains.

Comment 3. Line 128-130 - please clarify here the composition of the medium - ammonium is a preferred nitrogen source for Streptomyces, thus ensuring bacterial growth. Please clarify concentration of added nitrogen-containing compounds as well as the nature of used media.

Response: We are grateful for this relevant comment. Knowing that the ISP2 medium contains 10 g of malt extract and 4 g of yeast extract per liter, and that malt extract contains between 1.0 and 1.6% nitrogen and yeast extract between 9 and 11% nitrogen, the 14 g from these two sources provides approximately 0.5 g of nitrogen. Based on this information and calculations of the equivalent amount of nitrogen supplied by inorganic salts, it was established that 3.61 g of potassium nitrate, 2.36 g of ammonium sulfate, or 1.91 g of ammonium chloride were required as a nitrogen source while maintaining the other components of the medium. This is consistent with the information in Table 3, which contains the different levels of the independent variables evaluated in the model.

Comment 4. Line 165-166 - please clarify sample size and variation (SE or SD are not visible on the graph)

Response: We thank the reviewer for this observation. The figure referred to in this comment corresponds to a Pareto chart of standardized effects derived from the Plackett–Burman design, which identifies the relative influence of each tested factor on pigment production. In this type of chart, the bars represent the magnitude and statistical significance of each factor’s standardized effect, rather than mean values from replicated experimental data. Therefore, standard errors (SE) or standard deviations (SD) are not applicable in this representation. However, we clarified in the figure legend and in the Materials and Methods section that all experimental data used to calculate the standardized effects were obtained from biological triplicates (n = 3). Each factor’s effect value was computed according to the statistical model, and the chart indicates the hierarchy of factors' influence rather than variability between replicates.

Comment 5. Line 300 - Streptomyces as a genus name should be written in italics

Response: Thank you for this query. Streptomyces was corrected to “Streptomyces” in italics as cordially suggested.

Round 2

Reviewer 1 Report

Comments and Suggestions for Authors

its ok now and can be accepted.